# Assessment of hypertension and other factors associated with the severity of disease in COVID-19 pneumonia, Addis Ababa, Ethiopia: A case-control study

**Andargew Yohannes Ashamo**[1]◉*, **Abebaw Bekele**[2]‡, **Adane Petrose**[1], **Tsegaye Gebreyes**[2], **Eyob Kebede Etissa**[3], **Amsalu Bekele**[1], **Deborah Haisch**[4], **Neil W. Schluger**[5], **Hanan Yusuf**[1], **Tewodros Haile**[1], **Negussie Deyessa**[6]◉, **Dawit Kebede**[1,2]◉

1 Department of Internal Medicine, Division of Pulmonary and Critical Care, Addis Ababa University, Addis Ababa, Ethiopia, 2 Department of Internal Medicine, Eka Kotebe General Hospital, Addis Ababa, Ethiopia, 3 East African Training Initiative, Addis Ababa, Ethiopia, 4 Department of Internal Medicine, Weill Cornell Medical College, New York City, New York, United States of America, 5 Department of Internal Medicine, Westchester Medical Center, New York Medical College, New York City, New York, United States of America, 6 Department of Preventive Medicine, School of Public Health, Addis Ababa University, Addis Ababa, Ethiopia

◉ These authors contributed equally to this work.
‡ AB also contributed equally to this work.
* andargjhon@gmail.com

**Data Availability Statement:** All relevant data are within the manuscript.

## Abstract

### Background

Various reports suggested that pre-existing medical illnesses, including hypertension and other demographic, clinical, and laboratory factors, could pose an increased risk of disease severity and mortality among COVID-19 patients. This study aimed to assess the relation of hypertension and other factors to the severity of COVID-19 pneumonia in patients discharged from Eka Kotebe Hospital in June-September, 2020.

### Methods

This is a single-center case-control study of 265 adult patients discharged alive or dead, 75 with a course of severe COVID-19 for the cases arm and 190 with the non-severe disease for the control arm. Three age and sex-matched controls were selected randomly for each patient on the case arm. Chi-square, multivariable binary logistic regression, and odds ratio (OR) with a 95% confidence interval was used to assess the association between the various factors and the severity of the disease. A p-value of <0.05 is considered statistically significant.

### Results

Of the 265 study participants, 80% were male. The median age was 43 IQR(36–60) years. Both arms had similar demographic characteristics. Hypertension was strongly associated with the severity of COVID-19 pneumonia based on effect outcome adjustment (AOR =

**Funding:** The author(s) received no specific funding for this work.

**Competing interests:** The authors have declared that no competing interests exist.

2.93, 95% CI 1.489, 5.783, p-value = 0.002), similarly, having diabetes mellitus (AOR = 3.17, 95% CI 1.374, 7.313, p-value<0.007), chronic cardiac disease (AOR = 4.803, 95% CI 1.238–18.636, p<0.023), and an increase in a pulse rate (AOR = 1.041, 95% CI 1.017, 1.066, p-value = 0.001) were found to have a significant association with the severity of COVID-19 pneumonia.

## Conclusions

Hypertension was associated with the severity of COVID-19 pneumonia, and so were diabetes mellitus, chronic cardiac disease, and an increase in pulse rate.

## Introduction

Coronavirus disease 2019 (COVID-19) is a disease caused by the severe acute respiratory syndrome-coronavirus-2 (SARS-CoV-2) virus [1]. SARS-CoV-2 infection encompasses a clinical spectrum of asymptomatic infection, mild or moderate respiratory tract illness, and severe viral pneumonia with respiratory failure and even death [2]. The initial common clinical manifestations found in patients with COVID-19 included fever, cough, breathlessness, vomiting, and diarrhea [3]. The outcomes of asymptomatic to moderate COVID-19 are good, with most patients recovering with no sequelae [4]. By contrast, mortality from severe and critical cases may be as high as 50%, especially in the presence of end-organ damage and multiple-organ dysfunction syndrome [5]. Age above 65 and pre-existing medical illnesses, including hypertension, diabetes, cardiovascular disease, obesity, malignancy, chronic kidney disease, and chronic obstructive pulmonary disease, have been associated with increased morbidity and mortality [6, 7]. The prevalence of these factors in the population could help the potential burden of severe outcomes and stress on the health care system overall. Identifying clinical indicators of severe or fatal disease is necessary to enable risk stratification and optimize the allocation of limited resources.

Hypertension has been widely reported to increase the severity and mortality of patients with COVID-19. However, early COVID-19 studies have reported mixed findings concerning hypertension and other comorbidities. Findings from early studies on the association between hypertension and other comorbidities were not always adjusted for possible confounders, especially age, which has been the strongest predictor of risk for bad outcomes overall [8]. Overall, data have been inconclusive. Using logistic regression models adjusted for age and sex, some studies that conducted multivariate adjustments failed to document a significant association between hypertension and the severity of COVID-19 pneumonia [9]. On the contrary, in a larger study, hypertension was significantly associated with a 41% higher risk of mortality due to COVID-19 after adjusting for age, gender, comorbid diabetes, cerebrovascular diseases, and chronic renal disease using the Cox proportional hazard regression [10]. Of note, very few studies have examined risks for poor outcomes among persons infected with SARS-CoV-2 in sub-Saharan Africa, and in many African countries, including Ethiopia, the burden of the epidemic has appeared less than in many high-resource settings, including countries in North America and western Europe. The objective of our study was to assess the association of hypertension and other factors with the severity of COVID-19 pneumonia based on matched case-control among patients in Ethiopia, the second-largest country in Africa by population.

## Method

### Study area and period

This study was undertaken in Eka Kotebe General Hospital, which was the first and the only hospital in the capital, Addis Ababa, Ethiopia, dedicated as a whole to COVID-19 patients' isolation and treatment and it is still the only center that provides dialysis/renal replacement therapy, surgical, obstetrics and gynecologic care for COVID-19 positive patients which makes it the main center for referral and admission of cases who require advanced or critical care. It has a capacity of 600 beds, with 16 beds dedicated to intensive care services. Over 130 nurses, 90 general practitioners, two internists, one anesthesiologist, two emergency physicians, one pulmonology and critical care sub-specialist, two obstetrician and gynecologists, two surgeons, two psychiatrists, two radiologists, and two pediatricians were involved in the care of those patients. Seven of these senior physicians were academic staff at Addis Ababa University, College of Health Sciences, and they have been working in the hospital since April 2020. The data collection was conducted from September to November 2021.

### Source and study population

The source population was all patients admitted to the hospital with a laboratory-confirmed diagnosis of COVID-19 infection. The study population was all adult patients discharged during June-to-September 2020, before the availability of the COVID-19 vaccine at the peak of the first wave in the nation, from Eka Kotebe Hospital with severe/critical COVID-19 infection and their age (with five-year age band) and sex-matched control COVID-19 infected with mild/ moderate disease.

### Study design

The study design was a case-control study. Cases were all adult RT-PCR confirmed COVID-19 patients with severe/critical disease. At the same time, controls were those age and gender-matched RT-PCR confirmed COVID-19 patients with mild/moderate disease who were also hospitalized at that time.

### Inclusion criteria and exclusion criteria

Only those with symptomatic RT-PCR confirmed COVID-19 disease and discharged alive/ dead from June to September 2021 were included in the study. Cases were patients aged ≥18 with severe disease. In contrast, controls were age and sex-matched patients with mild/moderate disease. Patients with severe/critical COVID-19 for whom matched control with non-severe COVID-19 patient was not available, and patients for whom severity status and/or primary outcome not documented were excluded from this study.

### Sample size determination and sampling technique

The sample size was calculated using sample size determination for proportion in two populations using EPI-Info for case-control studies with an assumption of the two-sided significance level of 95%, power 80%, control to case ratio of 3 to 1, taking a proportion of hypertension among the controls of 19.6% [11] and with the assumption of the proportion to be double among the cases (39.2%). The calculated sample size was 240 using Fleiss with continuity correction. Adding 5% for incomplete data, the sample size became 252 with 63 patients on the case arm and 189 patients on the control arm. The study included all cases sequentially, and the next five consecutive non-cases for each case were included. Within the specified period of discharge times, we collected 265 patient data (75 cases and 190 controls).

## Data collection techniques

Standardized case report format (CRF) was used to collect enrolled patients' data on demographic, clinical diagnosis, laboratory, treatments, complications, and clinical outcomes from patients' medical records at Eka Kotebe hospital. For every severe/critical COVID-19 case enrolled for data collection, we enrolled controls to be those discharged (alive or dead) on the same or immediate five days. Two independent researchers carefully reviewed the data. Discrepancies between the reviewers were resolved by discussion. Data on all variables were collected by trained physicians working at Eka Kotebe Hospital using the CRF customized and prepared by the research team.

## Data quality and management

To ensure the quality of the data, training was given to data collectors in Addis Ababa for one day before the survey to ensure consistency and reduce Intra and inter-observer differences in the measurement of variables. The collected data were checked for completeness and consistency on each day of data collection. The assigned supervisors and principal investigators made supervision and monitoring every day.

## Data processing and analysis

After data collection, to ensure the quality of the data, the entire filled data were checked for incompleteness and inconsistency. The extracted data on the ODK-collect server were exported directly into SPSS version 25.0 for statistical analysis. Categorical variables were presented as frequency rates and percentages (%), and continuous variables were described using a median and interquartile range. We used a chi-square test to compare exposure variables between patients with severe disease and non-severe disease groups with categorical data. Based on the results of the univariate analysis, variables with a p-value < 0.3 were selected for the multivariable binary logistic regression model. The Hosmer and Lemeshow test (p = 0.112) was used to evaluate if the binary logistic regression model was fit. Multivariable logistic regression was used to assess the association between hypertension, other demographics, and clinical factors on the severity of COVID-19 pneumonia. The model included factors associated with or borderline associated with were included in the multivariable analysis. Crude and adjusted odds ratio (OR) displayed a 95% confidence interval association. P-value < 0.05 was considered statistically significant.

## Ethics approval and consent to participate

Ethical clearance was obtained from the AAU-CHS internal medicine department ethical committee and Eka Kotebe general hospital ethical review committee (approval letter with number 150/5/59). The study had no risk/negative consequence on those who participated in the study, and for the design of the study, we were not required to obtain informed consent from the participants. Medical record numbers were used for data collection, and personal identifiers were not used in the research report. The collected information was limited to the principal investigator, and confidentiality was maintained throughout.

## Operational definition

SARS-CoV-2 infection was confirmed after oropharyngeal swab samples were collected and tested using RT-PCR. Hypertension was defined according to Joint National Committee 8 guideline recommendation [12]. Hypertension (HTN), also known as high blood pressure (BP), was defined as BP ≥140/90 millimeters of mercury (mmHg). It was a measure by a

physician or other health care professional. Only those with an established diagnosis of hypertension at admission were included in this study. An average of five measurements was taken for each patient.

The clinical Spectrum of SARS-CoV-2 Infection was defined according to WHO and NIH COVID-19 Treatment Guideline [13, 14]. Patients with mild COVID-19 were those individuals who had any of the various signs and symptoms of COVID-19 (e.g., fever, cough, sore throat, malaise, headache, muscle pain, nausea, vomiting, diarrhea, loss of taste and smell) but who did not have shortness of breath, dyspnea, or abnormal chest imaging. Moderate COVID-19 included those with signs of lower respiratory disease during clinical assessment or imaging but who maintained a saturation of oxygen (SpO2) $\geq$94% on room air. Severe COVID-19 illness included individuals who had SpO2 <94% on room air at sea level, a ratio of the arterial partial pressure of oxygen to fraction of inspired oxygen (PaO2/FiO2) <300 mm Hg (SpO2/FiO2 = 315, according to Kigali definition), respiratory frequency >30 breaths/min, or infiltrates >50% of the lung parenchyma on chest imaging. Critical COVID-19 included individuals with respiratory failure, septic shock, and/ or multiple organ dysfunction. We relied on treating physician diagnosis in order to categorize patients with ARDS.

## Result

A total of 265 participants were included, 75 cases and 190 controls. Two hundred twelve (80%) were males and the majority (37.7%) were in the age group 40–59, with a median age of 43 IQR (36–60) (Table 1). A statistically significant difference in some of the vital signs was found using a chi-square test among the cases and controls. These were median higher pulse rate (89 bpm vs 86 bpm P<0.003), respiratory rate (27 bpm vs 20 bpm P < 0.000), systolic blood pressure (SBP) (124 mmHg vs 118 mmHg P < 0.006), and lower oxygen saturation (SpO$_2$) (93% vs 94% P < 0.0001) for those with severe or critical COVID-19 pneumonia (Table 2).

Pre-existing comorbidity was found in 35.8% of all the study participants, and 58.7% of those with severe or critical diseases. Hypertension, diabetes, and chronic cardiac and lung diseases were the most frequent comorbidities, with 19.6%, 11.3%, 4.2%, and 4.2%, respectively. COVID-19 patients with hypertension showed a significant association with severe disease (33.3%Vs14.2% p<0.01) compared with those with non-severe disease. Similarly, the presence of any comorbidity, diabetes, chronic cardiac disease, smoking, and malignancy, was all more frequent in the case group than in the control group (56.7% vs 26.8%, P<0.001; 21.3% vs 7.4% P = 0.001; 9.3% vs 2.1% P<0.05; 8.9% vs 1.5% P<0.05; and 4.0% vs 0.0% P<0.05). Regarding smoking only six (10.4%) participants reported as smokers, of which only 4 (8.9%) had severe disease, but there was no significant association with disease severity. Because there were 30.9% missed data for Body Mass Index (BMI), we could not include it in subsequent analysis.

Table 1. Demographic characteristics of study participants.

| Variables | | Severity | |
|---|---|---|---|
| | | Non-severe disease (n = 190) (%) | Severe disease(n = 75)(%) |
| Age(Median, IQR): | 20–39 | 68 (35.8) | 25 (33.3) |
| | 40–59 | 70 (36.8) | 30 (40.0) |
| | $\geq$60 | 52 (27.4) | 20 (26.7) |
| Sex | Male | 152 (80.0) | 60 (80.0) |
| | Female | 38 (20.0) | 15 (20.0) |

**Table 2. Admission vital signs profile of study participants.**

| Variables | Severity | | P-value |
|---|---|---|---|
| | Non-severe disease (n = 190) | Severe disease (n = 75) | |
| Temperature (Median, IQR) | 36.1 (36, 36.8) | 36.5 (36,37) | 0.068 |
| Pulse rate (Median, IQR) | 86 (79,90.2) | 89 (82, 100) | **0.003** |
| Respiratory rate (Median, IQR) | 20 (18, 22) | 27 (24, 33) | **0.000** |
| Systolic blood pressure (Median, IQR) | 118 (111,128) | 124 (115,136) | **0.006** |
| Diastolic blood pressure (Median, IQR) | 74 (70, 79) | 75 (70, 81) | 0.299 |
| Aterial saturation ($SpO_2$) (Median, IQR) | 94 (93, 95) | 93 (90, 94) | **0.000** |

However, analysis of the available data showed there was only one (0.4%) patient with BMI$\geq$30kg/m$^2$ and 35(13.2%) were overweight (BMI 25–29.9kg/m$^2$).

Fifteen (5.7%) were admitted directly to the ICU and eleven (4.2%) died from a course of severe disease while 254 (95.8%) were discharged alive (Table 3).

As shown in Table 4, there were some missing data for most laboratory findings where its percentage difference among the study groups and for laboratory tests ranging from 2.7% to 9.3% for the cases. However, after removing all missing data from each variable, the median value and proportion were compared with disease severity, and white cell (WBC) count, absolute lymphocyte count (ALC), Neutrophil-lymphocyte ratio (NLR), platelet count, Alanine Transaminase (ALT), Aspartate transaminase (AST), and urea were found to have significant association with disease severity. On the categorical analysis of the laboratory markers, those who had severe disease had significant leukocytosis ($>$ 10,000 cells/ µL) (30.1% vs 9.8% p $<$ 0.002); lymphopenia ($<$1000 cells/ µL) (63% vs 21.6% p $<$ 0.000); raised neutrophil-to-lymphocyte ratio of $\geq$ 10 (28.8% vs 8.8% p $<$ 0.001); thrombocytopenia ($<$150,000 cells/ µL) (16.4% vs 5% p $<$ 0.013); elevated ALT (35.7% vs 14.1% p $<$ 0.001); elevated AST (76.5% vs 28.1% p $<$ 0.0001); and elevated urea (31.9% vs 16.5% p $<$ 0.025) compared to those who had non-severe disease.

In addition to isolation and monitoring, different treatment armaments were used based on the severity and complications of the illness. Antibiotics, systemic anticoagulants, dexamethasone, prone positioning, vasopressor were most frequently used interventions, 40.4%, 31.3%, 29.1%, 19.6% and 2.6% respectively. For those with non-severe COVID-19 only antibiotics (16.8%) and anticoagulants (4%) were given from the list of interventions mentioned in Table 5. The rest of the interventions were given solely to those with severe diseases, indicating the interventions were not given or used in equal proportions in both groups. A therapeutic dose of anticoagulant was used in 36(13.6%) and experimental agents in 3(1.1%) patients, 2 were given hydroxychloroquine and 1 took remdesivir. Most observed complications were pneumonia in 24.4%, acute respiratory distress syndrome (ARDS) in 5.3%, acute kidney injury (AKI) in 4.9%, bacteremia in 3.4%, cardiac arrest in 3.4%, and shock in 3% of those with severe disease (Table 5).

Hypertension, pulse rate, respiratory rate, SBP, SpO2, comorbidity, diabetes, and chronic cardiac disease were all significantly associated with disease severity on univariate analysis. On the multivariable binary logistic regression analysis (adjusted odds ratio) with the omission of respiratory rate from this final analysis model, hypertension was associated with disease severity (AOR = 2.93, 95% CI 1.489, 5.783, p-value = 0.002). Furthermore, an increase in pulse rate, diabetes mellitus, and chronic cardiac disease were also significantly associated with COVID-19 pneumonia severity.

As the pulse rate increased by one beat per minute, the likelihood of the disease becoming severe increased by 1.04 times (AOR = 1.041, 95% CI 1.017, 1.066, p-value = 0.001). Again,

**Table 3. Distribution of comorbidities, disease severity, and outcome of study participants.**

| Variables | | Severity | | X² | P-value |
|---|---|---|---|---|---|
| | | Non-severe disease (n = 190) | Severe disease (n = 75) | | |
| Comorbidity | Yes | 51 (26.8) | 44 (58.7) | 23.6 | **0.000** |
| | No | 139 (73.2) | 31 (18.2) | | |
| Hypertension | Yes | 27 (14.2) | 25 (33.3) | 12.4 | **0.000** |
| | No | 163 (85.8) | 50 (66.7) | | |
| Diabetic mellitus | Yes | 14 (7.4) | 16 (21.3) | 10.4 | **0.001** |
| | No | 176 (92.6) | 59 (78.7) | | |
| Chronic pulmonary disease | Yes | 8 (4.2) | 3 (4.0) | 0.0 | 1.000 |
| | No | 182 (95.8) | 72 (96.0) | | |
| Chronic cardiac disease | Yes | 4 (2.1) | 7 (9.3) | 7.0 | **0.014** |
| | No | 186 (97.9) | 68 (90.7) | | |
| HIV[a] | Yes | 6 (3.2) | 4 (5.3) | 0.7 | 0.585 |
| | No | 182 (95.8) | 70 (93.3) | | |
| | Unknown | 2 (1.1) | 1 (1.3) | | |
| Smoking | Yes | 2 (1.5) | 4 (8.9) | 5.6 | **0.018** |
| | No | 131 (98.5) | 41 (91.1) | | |
| Tuberculosis | Yes | 2 (1.1) | 2 (2.7) | 0.9 | 0.318 |
| | No | 188 (98.9) | 73 (97.3) | | |
| Malignancy | Yes | 0 (0.0) | 3 (4.0) | 7.6 | **0.022** |
| | No | 190 (100.0) | 72 (96.0) | | |
| CKD[b] | Yes | 0 (0.0) | 2 (2.7) | 5.1 | 0.079 |
| | No | 190 (100.0) | 73 (97.3) | | |
| Direct ICU admission | Yes | 0 (0.0) | 15 (20) | 40.2 | **0.000** |
| | No | 190 (100.0) | 60 (80.0) | | |
| Outcome | Discharged | 190 (100.0) | 64 (85.3) | 29.0 | **0.000** |
| | Death | 0 (0.0) | 11 (14.7) | | |

a-Human immunodeficiency virus

b-Chronic kidney disease.

patients with diabetes have a 3.1-fold increased likelihood of having the severe disease when compared to non-diabetic patients (AOR = 3.17, 95% CI 1.374, 7.313, p-value<0.007), and those with chronic cardiac disease 4.8-fold increased tendency to have severe disease (AOR = 4.803, 95% CI 1.238–18.636, p<0.023) (Table 6).

## Discussion

We assessed the association between the severity of COVID-19 pneumonia and the demographic, clinical, and laboratory profile of discharged alive or dead patients. The chi-square result showed a significant difference in vital signs and comorbidities among patients with non-severe and severe diseases. This association showed that the severe disease group had had higher respiratory rate, pulse rate, systolic blood pressure, lower SPO2, one or more comorbid illnesses, had hypertension, diabetes mellitus, chronic cardiac and pulmonary diseases; and, lower ALC, higher NLR, relatively lower platelet count and raised ALT, AST, and urea. This change in vital status suggests the possible contribution of demographic and clinical characteristics to disease severity.

The univariate analysis has shown pulse rate, respiratory rate, SBP, SpO2, presence of any comorbidity, hypertension, diabetes mellitus, and chronic cardiac disease were significantly

**Table 4. Distribution of laboratory values and disease severity of study participants.**

| Variables | | Severity | | $X^2$ | P-value |
|---|---|---|---|---|---|
| | | Non-severe disease | Severe disease | | |
| Haemoglobin g/dl (Median, IQR) n = 173 | | 14.3 (13,15.8) | 14 (12.8, 15.1) | | 0.196 |
| | <12 | 7 (3.7) | 10(14.1) | 2.4 | 1.16 |
| | ≥12 | 95 (93.1) | 61 (85.9) | | |
| WBC[a] $10^3$ count cells/μL (Median, IQR) n = 175 | | 5.83 (5.14,7.19) | 7.74 (5.38,10.6) | | **0.000** |
| | <4 | 11 (10.8) | 4 (5.5) | 12.3 | **0.002** |
| | 4–10 | 81 (79.4) | 47 (64.4) | | |
| | ≥ 10 | 10 (9.8) | 22 (30.1) | | |
| ALC[b] 103 count cells/μL (Median, IQR) n = 175 | | 1400 (1088.2,1643) | 845 (535,1222.2) | | **0.000** |
| | < 1000 | 22 (21.6) | 46 (63.0) | 30.7 | **0.000** |
| | ≥ 1000 | 80 (78.4) | 27 (37.0) | | |
| NLR[c] (Median, IQR) n = 175 | | 3 (2.0,3.2) | 7.2 (3.8,11.4) | | **0.000** |
| | < 10 | 93 (91.2) | 52 (71.2) | 11.9 | **0.001** |
| | ≥ 10 | 9 (8.8) | 21 (28.8) | | |
| Platelet (Median, IQR) n = 173 | | 223,500 (191,250–279,000) | 223,000 (160,000–335,500) | | 0.972 |
| | Yes | 5 (5.0) | 12 (16.4) | 6.2 | **0.013** |
| | No | 95 (95.0) | 61 (83.6) | | |
| ALT[d] (Median, IQR) n = 162 | | 29 (22, 41) | 42 (34.7, 74.7) | | **0.000** |
| | < 60 | 79 (85.9) | 45 (64.3) | 10.3 | **0.001** |
| | ≥ 60 | 13 (14.1) | 25 (35.7) | | |
| AST[e] (Median, IQR) n = 157 | | 25 (20, 38) | 51.5 (38, 62.5) | | **0.000** |
| | < 37 | 64 (71.9) | 16 (23.5) | 36.1 | **0.000** |
| | ≥ 37 | 25 (28.1) | 52 (76.5) | | |
| Urea (Median, IQR) n = 154 | | 13 (10, 17) | 17 (12, 23) | | **0.003** |
| | < 20 | 71 (83.5) | 47 (68.1) | 5.0 | **0.025** |
| | ≥ 20 | 14 (16.5) | 22 (31.9) | | |
| Creatinine (Median, IQR) n = 162 | | 0.83 (0.68, 1.04) | 0.87 (0.74, 1.02) | | 0.113 |
| | < 1.2 | 85 (93.4) | 61 (85.9) | 2.5 | |
| | ≥ 1.2 | 6 (6.6) | 10 (14.1) | | |

a-WBC-white cell count,

b-Absolute lymphocyte count,

c-Neutrophil lymphocyte count,

d-Alanintransaminase,

e-Aspartatetransaminase.

associated with the severity of the disease, indicating that these factors independently contributed to the severity of COVID-19 pneumonia. After controlling other covariates upon further analysis using multivariate binary logistic regression, especially removing the respiratory rate from the final model for its mediator effect on disease severity (upon assessment for bivariate collerate and multicollinear effect, respiratory rate was the only variable with a strong modifying effect on the association of hypertension and other factors on the severity of COVID-19 pneumonia), hypertension showed a significant association to disease severity, as it has shown a significant association on univariate logistic regression. Hypertension was the most prevalent comorbidity in COVID-19 patients; estimated prevalence rates ranged from 15.0% to 36.5%

**Table 5. Treatment and complications of study participants.**

| Variables | | n (%) | Variables | | n (%) |
|---|---|---|---|---|---|
| IV fluid | Yes | 21 (7.9) | Complications | | |
| | No | 244 (92.1) | Shock | Yes | 8 (3.0) |
| Corticosteroids | Yes | 75 (28.3) | | No | 257 (97.0) |
| | No | 190 (71.7) | Meningitis/Encepha litis | Yes | 2 (0.8) |
| Antibiotics | Yes | 107 (40.4) | | No | 262 (98.9) |
| | No | 158 (59.6) | Cardiac arrhythmia | Yes | 6 (2.3) |
| Antifungal | Yes | 4 (1.5) | | No | 259 (97.7) |
| | No | 261 (98.5) | Cardiac arrest | Yes | 9 (3.4) |
| Experimental agent | Yes | 3 (1.1) | | No | 256 (96.6) |
| | No | 262 (98.9) | Pneumonia | Yes | 64 (24.2) |
| Systemic anti-coagulant | Yes | 83 (31.3) | | No | 201 (75.8) |
| | No | 182 (68.7) | ARDS | Yes | 14 (5.3) |
| Patters of anti-coagulant | Therapeutic | 36 (13.6) | | No | 251 (94.7) |
| | Prophylactic | 47 (17.7) | Bacteraemia | Yes | 9 (3.4) |
| Oxygen therapy | Yes | 75 (28.3) | | No | 256 (96.6) |
| | No | 190 (71.7) | Acute renal injury | Yes | 13 (4.9) |
| Prone | Yes | 52 (19.6) | | No | 252 (95.1) |
| | No | 7 (2.6) | Liver dysfunction | Yes | 5 (1.9) |
| | Unspecified | 16 (6.0) | | No | 260 (98.1) |
| Vasopressors | Yes | 7 (2.6) | Cardiomyopathy | Yes | 1 (0.4) |
| | No | 258 (79.4) | | No | 264 (99.6) |

[9, 15]. Some associations between hypertension and the severity of the COVID-19 illness were suggested earlier during the pandemic. HE Abraha et al., Jean B. Nachega et al., Guan WJ et al., and Wang D et al. reported a 24.6%, 24.5%, 23.7%, and 58.3% prevalence rate of hypertension among patients with severe COVID-19 pneumonia, respectively [1, 12, 16, 17].

Our results are consistent with systematic reviews and meta-analyses indicating that hypertension was the most prevalent chronic morbidity in COVID-19 patients [17%; 95% confidence interval (CI):14–22%]. One meta-analysis showed that the odds (OR) of hypertension in patients with severe disease, in comparison to those with non-severe disease, was 3.42 (95%CI: 1.88–6.22) [18, 19]. One of the early pieces of evidence during the pandemic, a retrospective

**Table 6. Determinants of disease severity.**

| Variables | COR (95% CI) | P-value | AOR* (95% CI) | p-value |
|---|---|---|---|---|
| Pulse rate | 1.036 (1.013–1.060) | 0.002 | 1.041 (1.017–1.066) | **0.001** |
| Hypertension | | | | |
| Yes | 3.019 (1.608–5.665) | 0.001 | 2.934 (1.489–5.783) | **0.002** |
| No | 1 | | 1 | |
| Diabetes Mellitus | | | | |
| Yes | 3.409 (1.570–7.404) | 0.002 | 3.170 (1.374–7.313) | **0.007** |
| No | 1 | | 1 | |
| Chronic cardiac disease | | | | |
| Yes | 4.787 (1.358–16.867) | 0.015 | 4.803 (1.238–18.636) | **0.023** |
| No | 1 | | 1 | |

*Adjusted odds ratio(AOR) after the respiratory rate was removed or omitted from the final model for its mediator effect.

case study from china, showed that a quarter of patients had at least one comorbidity. The most prevalent comorbidity was hypertension (16.9%), followed by diabetes (8.2%), with an increased risk for severe disease by 1.6-fold and 1.6-fold, respectively. Those with two or more comorbidities were 2.5 times at risk for a severe and fatal course of the disease [19].

In one particular retrospective study in Lagos, Nigeria with a total of 2075 adult COVID-19 cases, the prevalence of hypertension was 17.8% followed by diabetes (7.2%). Hypertension posed an increased risk (approx. 4-fold) of severe COVID-19 in the presence of multiple comorbidities [7]. A meta-analysis also showed that hypertension was significantly associated, nearly 1.5 times, with the increased risk of adverse outcomes in COVID-19 patients. The sub-group analysis with an adjusted odds ratio showed a significant correlation between hypertension and adverse outcomes [20]. From the pooled analysis performed on 13 studies with a total of 2893 COVID-19 patients hypertension was found to be associated with a nearly 2.5-fold significantly enhanced risk of severe COVID-19 disease [21]. Because the association of hypertension with the adverse outcomes of COVID-19 patients might be affected by various factors such as age, gender, and other comorbidities, it is recommended to have adjusted effect estimates for confounders before we concluded which we did in our study.

Similarly, diabetes mellitus, chronic cardiac disease, and pulse rate were also factors significantly associated with the severity of the disease. Having diabetes mellitus was found to be an important predictor of disease severity. Especially if the diabetes is poorly controlled, it is known to lead to compromised immunity that decreases the body's ability to clear infections, including SARS-CoV2 infection. And has a high chance of having additional chronic illness than non-diabetic patients. Diabetes mellitus has rapidly become established as a major comorbidity for severe COVID-19 disease. It is believed that diabetic individuals are not more susceptible to developing viral infection per se but are more likely to show increased clinical severity and to require ICU admission. Body mass index was independently associated with adverse outcomes in COVID-19 patients with diabetes [22]. Hyperglycemia, altered immune function, sub-optimal glycaemic control during admission, reduced forced vital capacity, and pro-thrombotic and pro-inflammatory state makes them vulnerable to severe and critical illness with complications [23, 24]. This is consistent with reports of association with severe COVID-19 pneumonia and poor prognosis from other counties [25–28] and local studies [29].

In a meta-analysis of eight studies, cardiovascular disease was one of the most prevalent comorbidities following hypertension and diabetes [18], also shown in a meta-analysis by Wang et al. [30] and Huang et al. [2]. Compared to patients with non-severe disease, the pooled odds ratio of cardiovascular disease in severe cases was 3.42, closer to our study finding [18]. From the first systematic review and meta-analysis focusing on the relationship of severe COVID-19 with cardiovascular disease and its risk factors, the majority of studies showed a positive association between prior chronic cardiac disease and severe COVID-19, with the primary pooled relative risk of 5.05 and the pooled adjusted relative risk estimate was 1.75, where it is accounted for confounders especially age [31].

The explanation for the higher prevalence of hypertension, diabetes mellitus, and cardiac disease among COVID-19 patients may focus on the SARS-CoV-2 cell entry mechanism. Similar to SARS-CoV-1, SARS-CoV-2 contains a receptor-binding domain (RBD) that recognizes angiotensin-converting enzyme 2 (ACE2) as its receptor with a higher binding affinity compared to SARS-CoV-1 [32]. Epithelial cells of the lungs, intestine, kidney, and blood vessels are identified to have abundant ACE2 receptors [33]. Hence, increased expression of ACE2 may promote the internalization of SARS-CoV-2, which in turn may increase the chances of developing COVID-19 or a severe form of the disease [34]. The severity of COVID-19 among patients with hypertension, diabetes, and chronic cardiac disease also may be partially explained by the increased incidence of thrombotic complications which is a fact that patients

with these conditions are at increased risk of thrombotic events [35, 36]. These chronic medical conditions often present with inflammation and weakened innate immune responses in affected individuals. This may predispose those individuals to infections and disease complications [18]. The high prevalence of fatal cases among COVID-19 patients with hypertension, diabetes, and chronic cardiac disease as comorbidity could be due to the induction of cytokine storm. Cytokine storms resulting in hyper-inflammation are the hallmarks of severe SARS-CoV-2 infection [37]. Metabolic inflammation as a consequence of hypertension, diabetes, and chronic cardiac disease is also known to compromise the immune system. These patients are commonly reported to have weakened immunological function arising from reduced macrophage and lymphocyte activity which could predispose individuals to infections, especially those infections for which cell-mediated immunity constitutes an important host defense [38].

Furthermore, the study showed that a single increase in pulse rate is one of the factors for having severe or critical COVID-19 pneumonia. An increase in pulse rate indicates a cardiorespiratory or metabolic disease, resulting in a poor prognosis. An increase in heart rate, and even more variability, have been related to worse outcomes in infection [39]. The therapeutic trial to lower it in septic patients has not been associated with an improvement in cardiac function [40] nor with the amelioration of mortality risk [41]. From this evidence, one could speculate that heart rate in infection is simply a marker of a severe clinical condition and a response to sepsis at presentation. However, others believe it could be related to the emergence of autonomic dysfunction [42]. In a patient with COVID-19, the persistence of increased heart rate has been assumed to be related to a dysregulation of an autonomic system [43]. A heart rate higher than 100 beats per minute has also been inserted into the latest European guidelines on hypertension as a prognostic marker since it is related to future cardiovascular events [44]. Discharge heart rate is strongly related to the evidence of a severe disease defined as the need for ICU admission and/or mechanical ventilation. The increase in heart rate at discharge was almost four times higher in patients with severe disease than in patients without severe disease (15.2% vs. 4%) [45]. Some clinicians are interested more in the persistence of sinus tachycardia over time with symptoms that could remain for longer than 3 weeks [46].

The other relevant factors identified in most studies are deranged laboratory parameters of WBC, ALC, NLR, platelet count, ALT, AST, and urea, [26, 27, 47, 48]. Though the data were incomplete for some of our cases and controls, it had shown an association with the disease severity on the chi-square test.

Most notably, similar to other local and regional [7, 17] studies our findings showed that the majority of study participants including those with severe diseases were young, however, only nearly one-third of the participants had comorbidity which was slightly lower than other local studies [29]. Interestingly, lower mortality rates are reported in most African countries compared to the global trend, unlike in Europe and the USA [49–52]. Though the reason is not fully understood, it is postulated that it could be related to warmer weather and the predominant youth population [52]. However, our finding showed an association between some comorbidities and disease severity among Ethiopian patients which could be also the case in other African patients as there is an epidemiological transition in a rise of non-communicable diseases in sub-Saharan Africa [53]. Earlier reports from the UK showed increased mortality among ethnic minorities where the exact explanations were not clear, but it was believed to be due to biological, medical, or sociological factors [54, 55]. Given the known risk factors for COVID-19 complications, the confluence of hypertension, diabetes, obesity, and the higher prevalence of cardiovascular disease among black persons may be driving these early signals. Among the two most populous countries in Africa, Nigeria and Ethiopia, the national prevalence of hypertension is 28.9% and 19.6% and diabetes mellitus is 5.77% and 5%, respectively [11, 56–58]. And where there is a lack of awareness and poor health-seeking behavior in our

population, many patients present with uncontrolled diseases with attending complications which could contribute to the severity of their illness [59].

## Strength and limitation

The main strength of this study was the enrolment of patients with COVID-19 confirmed by RT-PCR from a dedicated treatment center where the admission protocol and case management were consistent. It was also a matched study with an enrollment of subsequent cases for the controls discharged in the nearest one week. However, this study had a few limitations. First, the strict matching criteria, including the date of discharge on selecting one case for three controls, failed to fulfill the planned case-to-control ratio, we collected data for 75 cases for which we were supposed to collect proportionally 225 controls which makes the total sample size to be 300. However, we were able to retrieve data of 75 cases for 190 controls which made the total sample size 265, which is a bit higher than the initial calculated sample size (252, 63 cases for 189 controls) with a slightly higher number of those on case arm (75 vs 63). Furthermore, data were missing due to the study's retrospective nature, thus affecting detailed analyses of factors that may impact the severity of the disease. And thus, incomplete data on vital signs, behavioral and comorbidity documentation, and key laboratory tests limited the power of association analyses. In addition, the number of patients with some specific comorbidities was too small to conclude that such diseases, particularly CKD and malignancy, do not influence the severity of COVID-1.

## Conclusion

Hypertension was strongly associated with the severity of COVID-19 pneumonia based on effect outcome adjustment. And also, an increase in pulse rate, diabetes mellitus, and chronic cardiac disease was strongly associated with the severity of the disease. A single increase in pulse rate should alert the triage or treating team for early anticipation of a high chance of progressing to severe or critical disease and close monitoring for pulse rate. We also advise strict infection prevention and a low threshold for early detection, management, and anticipation of adverse outcomes among patients with hypertension, diabetes mellitus, and chronic cardiac disease.

## Acknowledgments

We would like to thank Eka Kotebe general hospital for facilitating the research work.

## Author Contributions

**Conceptualization:** Andargew Yohannes Ashamo, Negussie Deyessa, Dawit Kebede.

**Data curation:** Andargew Yohannes Ashamo, Abebaw Bekele, Adane Petrose, Tsegaye Gebreyes, Negussie Deyessa.

**Formal analysis:** Andargew Yohannes Ashamo, Eyob Kebede Etissa, Negussie Deyessa.

**Methodology:** Andargew Yohannes Ashamo, Tewodros Haile, Negussie Deyessa, Dawit Kebede.

**Project administration:** Abebaw Bekele.

**Supervision:** Andargew Yohannes Ashamo, Abebaw Bekele, Adane Petrose, Tsegaye Gebreyes, Tewodros Haile, Negussie Deyessa.

**Writing – original draft:** Andargew Yohannes Ashamo, Eyob Kebede Etissa.

**Writing – review & editing:** Amsalu Bekele, Deborah Haisch, Neil W. Schluger, Hanan Yusuf,
Tewodros Haile, Negussie Deyessa, Dawit Kebede.

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
