## [Decision Letter · Decision Letter 0]

15 Mar 2022

PONE-D-22-04541Assessment of Hypertension and Other Factors Associated with Severity of Disease in COVID-19 Pneumonia, Addis Ababa, Ethiopia: A Case-Control StudyPLOS ONE

Dear Dr. Ashamo,

Thank you for submitting your manuscript to PLOS ONE. After careful consideration, we feel that it has merit but does not fully meet PLOS ONE’s publication criteria as it currently stands. Therefore, we invite you to submit a revised version of the manuscript that addresses the points raised during the review process.

We look forward to receiving your revised manuscript.

Kind regards,

Kanhaiya Singh, Ph.D

Academic Editor

PLOS ONE

Journal Requirements:

2. Please amend either the title on the online submission form (via Edit Submission) or the title in the manuscript so that they are identical.

Additional Editor Comments:

Although reviewers have found this study interesting they have raised some significant questions. The major concern is about the novelty of this study as raised by Reviewer 3. In addition to addressing other comments raised by reviewers, please specifically and categorically explain in the manuscript about how this study is different than what published by Leulseged et al, 2022.

Reviewers' comments:

Reviewer's Responses to Questions

**Comments to the Author**

1. Is the manuscript technically sound, and do the data support the conclusions?

Reviewer #1: Yes

Reviewer #2: Yes

Reviewer #3: Yes

2. Has the statistical analysis been performed appropriately and rigorously? 

Reviewer #1: Yes

Reviewer #2: Yes

Reviewer #3: Yes

3. Have the authors made all data underlying the findings in their manuscript fully available?

Reviewer #1: Yes

Reviewer #2: Yes

Reviewer #3: Yes

4. Is the manuscript presented in an intelligible fashion and written in standard English?

Reviewer #1: Yes

Reviewer #2: Yes

Reviewer #3: No

5. Review Comments to the Author

Reviewer #1: This is an excellent report based on very thorough research. The members of research team ensured that the manuscript adheres to high ethical standards. Overall, this is a concise and well-written manuscript. The introduction is relevant and theory-based. The authors have included sufficient information about the previous studies in the discussion section, which will aid readers in following the rationale and procedures of the present study. The data of the manuscript contributes in understanding of COVID-19 related comorbidities and thy significant for understanding COVID associated disease aetiology.

However, the following points should be considered in the present form of this manuscript.

• In result section of the manuscript, the authors have reported positive association between many clinical characteristics and COVID associated Comorbidities. In discussion section the authors have mention about the findings. In my view authors should discuss about the possible pathological features by which mentioned clinical characteristics can contribute in disease severity. This will aid readers to follow the rationale of the study. (Note: Authors should to relate the results from the studies referring SARS-Cov and MERS-CoV which are structurally similar to COVID19 virus. Just reference to study which are associates demographic and clinical characteristics with viral disease severity should be good enough)

• In my view, authors should also consider applicability of Cox proportional hazard model (CH Model) for the association of various clinical indices and vital signs with the COVID abide severity because of following points:

CH model Adjusts multiple risk factors simultaneously.

• CH model also aid to limit the number of strata.

• Provides estimates and confidence intervals of how the risk changes (Clinical characteristics) across the strata and across unit increases the quantitative variables

• The severity of the viral disease also showed significant differences based on ethnicity and demography of the population, In my view, authors have reported associated between various clinical indices and disease severity in a African population. This may aid in understanding physiological parameter which resulted in heavy COVID severity in the African population and also hold population centric translational significance. So, I would suggest authors to add this crucial suggestion in discussion part of the manuscript.

Minor comments:

• Authors should rewrite the results section to ensure the result section looks well connected and free from any grammatical errors.

• Please sure that ethical protocol approval number and certificate should be incorporated in the supplementary data alongside a sample consent form approved by the ethical committee.

Reviewer #2: The data presented in the manuscript is collected from the patients admitted with severe/non-severe COVID-19 disease at the Eka Kotebe General Hospital, COVID-19 Isolation and Treatment Center. The treatment regimen remained standard within the study group. The manuscript is well written with statistically significant evaluation of the comorbidities associated with COVID-19.

Reviewer #3: This is an association study between different risk factors with the severity of COVID-19 pneumonia in patients. The study included 265 severe COVID-19 and 190 with the non-severe disease. While the data is interesting, the manuscript has significant omissions. My primary concerns are the following:

- Recent publication from Leulseged et al, 2022 “COVID-19 disease severity and associated factors among Ethiopian patients: A study of the millennium COVID-19 care center” have performed similar study in Ethiopian population in 686 patients. Can authors address how their study is different from this recent publication?

- The key problem with this study is that in the complex phenotype field, few researchers would have confidence in an association that is based on these small sample sizes. (Even if the authors stated as the power of the study).

- The discussion could be made much more interesting. As it is, they primarily reiterate their results and provide a bit more literature review. However, it would be interesting if they could discuss clinical implications and areas for further study.

6. PLOS authors have the option to publish the peer review history of their article (what does this mean?). If published, this will include your full peer review and any attached files.

Reviewer #1: No

Reviewer #2: No

Reviewer #3: No

---

## [Author Response · Author response to Decision Letter 0]

22 Apr 2022

Academic editor comments

Journal Requirements:

 Response: Thank you for the comment and we edited the manuscript as to PLOS ONE journal requirements. The revisions we made are:

a) The title, headings, and subsection were edited to sentence case with their respective level font size. 

b) The affiliation of the authors was corrected as to the guideline; department and city were included.

c) We removed the authors' declaration, contribution, funding, and conflicts of interest section, as it is well illustrated with the track change. 

d) The ethics statement is moved to the Methods section

e) The references are cited in the main text in brackets and are edited according to the journal guidelines, for references with more than six authors the first six author names are listed followed by “et al.”

2. Please amend either the title on the online submission form (via Edit Submission) or the title in the manuscript so that they are identical.

 Response: Thank you for the suggestion.

Full title from online submission form: ‘Assessment of Hypertension and Other Factors Associated with Severity of Disease in COVID-19 Pneumonia, Addis Ababa, Ethiopia: A Case-Control Study’ (Short Title: Risk Factors associated with Severity of COVID-19)

Title in the manuscript: ‘Assessment of hypertension and other factors associated with the severity of disease in COVID-19 pneumonia, Addis Ababa, Ethiopia: a case-control study ’

Apart from the edition of title to sentence case for the title in the manuscript, there was the omission of an article ‘the’ from the one written on the online submission. So, we amend that the title in the manuscript is the correct one and we will also edit accordingly.

 Response: Thank you for the comment and it is well amended. It is removed and incorporated in the Methods section accordingly (as depicted with track change)

Additional Editor Comments:

Although reviewers have found this study interesting they have raised some significant questions. The major concern is about the novelty of this study as raised by Reviewer 3. In addition to addressing other comments raised by reviewers, please specifically and categorically explain in the manuscript about how this study is different than what published by Leulseged et al, 2022.

 Response: Thanks for the concern raised. 

We are aware of the recent report from the Millennium Center, which has conclusions that are similar to our study. However, we feel that our paper adds to the literature in a significant manner. Ethiopia is the second-largest country in Africa by population, with some 115 million people, and a single report from a single center simply cannot suffice to describe any aspect of Covid-19 completely. Our study reports a cohort independent from the prior study, with a different methodology. Even if the results are somewhat similar, these are valuable and novel data.

Our study aimed to show mainly the association between hypertension and severity of COVID-19 pneumonia, and we also assessed the relation with other comorbidities, clinical, and laboratory characteristics of patients admitted at Eka General Hospital, which was the first and the only hospital in the capital dedicated as a whole for COVID-19 patients isolation and treatment and it is still the only center that provides dialysis/renal replacement therapy, surgical, obstetrics and gynecologic care for COVID-19 positive patients which makes it the main center for referral and admission of cases who require advanced or critical care. We compared differences between those with non-severe (mild/moderate COVID-19) to those with severe (severe/critical COVID-19) diseases, which we considered a matched case-control design to show an association. Whereas, Leulseged et al, 2022 study assessed COVID-19 patients admitted to Millunium COVID-19 Care Center (MCCC), the other care center in the capital which was a repurposed field hospital from a multipurpose recreation/exhibition center, for any characteristics associated with COVID-19 severity. They conducted a cross-sectional study on all selected patients of all age groups. Regarding ours, it was a matched case-control study, where cases were all adult RT-PCR confirmed COVID-19 patients with severe/critical disease. At the same time, controls were those age and gender-matched RT-PCR confirmed COVID-19 patients with mild/moderate disease who were discharged within the study period.

Furthermore, the other difference was sample size determination and sampling technique, they determined it using a double population proportion formula with the assumptions of; a 95% confidence interval, power of 80%, the proportion of males who had the severe disease as 80%, the proportion of females who had the severe disease as 75% and considering a non-response rate of 10%, using a simple random sampling technique. In our study, the sample size determination was based on the assumption of the two-sided significance level of a 95%, power of 80%, control to case ratio of 3 to 1, taking a proportion of hypertension among the controls to be 19.6% (inferred from the population-based study) and with the assumption of the proportion to be double among the cases (39.2%). The study included all cases sequentially, and the next five consecutive non-cases for each case were included (which were age and gender-matched).

Despite the above difference, we learned a lot from their study and tried to improve some aspects of our association study. Notably, we decided to match the participants for important demographic characteristics and to omit clinical characteristics that had pathobiological or effect-modifying relation with a severe manifestation of a disease (in our case we omitted such potential modifiers after checking for bivariate collerate and multicollinear effect).

In our study, the association assessment was focused on a specific factor or variable mainly (hypertension) as an entry with patients grouped as severe and non-severe, whereas the Leulseged et al. 2022 tried to show association assessment with several factors across the severity spectrum of COVID-19 (mild, moderate, severe) separately in addition to the difference in the study population, sample size determination, and sampling technique.

Reviewers' comments:

Reviewer #1: This is an excellent report based on very thorough research. The members of research team ensured that the manuscript adheres to high ethical standards. Overall, this is a concise and well-written manuscript. The introduction is relevant and theory-based. The authors have included sufficient information about the previous studies in the discussion section, which will aid readers in following the rationale and procedures of the present study. The data of the manuscript contributes in understanding of COVID-19 related comorbidities and thy significant for understanding COVID associated disease aetiology.

However, the following points should be considered in the present form of this manuscript.

1)• In result section of the manuscript, the authors have reported positive association between many clinical characteristics and COVID associated Comorbidities. In discussion section the authors have mention about the findings. In my view authors should discuss about the possible pathological features by which mentioned clinical characteristics can contribute in disease severity. This will aid readers to follow the rationale of the study. (Note: Authors should to relate the results from the studies referring SARS-Cov and MERS-CoV which are structurally similar to COVID19 virus. Just reference to study which are associates demographic and clinical characteristics with viral disease severity should be good enough)

 Response: Thank you, that is important and incorporated in the discussion section.

 -In paragraph 7 of the Discussion section:

 The explanation for the higher prevalence of hypertension, diabetes mellitus, and cardiac disease among COVID-19 patients may focus on the SARS-CoV-2 cell entry mechanism. Similar to SARS-CoV-1, SARS-CoV-2 contains a receptor-binding domain (RBD) that recognizes angiotensin-converting enzyme 2 (ACE2) as its receptor with a higher binding affinity compared to SARS-CoV-1 [32]. Epithelial cells of the lungs, intestine, kidney, and blood vessels are identified to have abundant ACE2 receptors [33]. Hence, increased expression of ACE2 may promote the internalization of SARS-CoV-2, which in turn may increase the chances of developing COVID-19 or a severe form of the disease[34]. The severity of COVID-19 among patients with hypertension, diabetes, and chronic cardiac disease also may be partially explained by the increased incidence of thrombotic complications which is a fact that patients with these conditions are at increased risk of thrombotic events [35,36]. These chronic medical conditions often present with inflammation and weakened innate immune responses in affected individuals. This may predispose those individuals to infections and disease complications [37]. The high prevalence of fatal cases among COVID-19 patients with hypertension, diabetes, and chronic cardiac disease as comorbidity could be due to the induction of cytokine storm. Cytokine storms resulting in hyper-inflammation are the hallmarks of severe SARS-CoV-2 infection [38]. Metabolic inflammation as a consequence of hypertension, diabetes, and chronic cardiac disease is also known to compromise the immune system. These patients are commonly reported to have weakened immunological function arising from reduced macrophage and lymphocyte activity which could predispose individuals to infections, especially those infections for which cell-mediated immunity constitutes an important host defense [39].

2)• In my view, authors should also consider applicability of Cox proportional hazard model (CH Model) for the association of various clinical indices and vital signs with the COVID abide severity because of following points:

CH model Adjusts multiple risk factors simultaneously.

• CH model also aid to limit the number of strata.

• Provides estimates and confidence intervals of how the risk changes (Clinical characteristics) across the strata and across unit increases the quantitative variables

 Response: Thank you for the comment and suggestion. Because we were primarily interested in the effect of hypertension, a widely reported risk for poor outcomes in patients with Covid-19 illness, we chose to perform multivariable binary regression. We agree with the reviewer that using a Cox proportional hazards approach could also be informative for looking at a multiplicity of risks, and we will perhaps plan to perform such an analysis in a later study

3)• The severity of the viral disease also showed significant differences based on ethnicity and demography of the population, In my view, authors have reported associated between various clinical indices and disease severity in a African population. This may aid in understanding physiological parameter which resulted in heavy COVID severity in the African population and also hold population centric translational significance. So, I would suggest authors to add this crucial suggestion in discussion part of the manuscript.

 Response: Thank you for this vital comment and it is incorporated.

In paragraph 10 of the Discussion section:

 Interestingly, lower mortality rates are reported in most African countries compared to the global trend, unlike in Europe and the USA [50,51,52,53]. Though the reason is not fully understood, it is postulated that it could be related to warmer weather and the predominant youth population[53]. However, our finding showed an association between some comorbidities and disease severity among Ethiopian patients which could be also the case in other African patients as there is an epidemiological transition in a rise of non-communicable diseases in sub-Saharan Africa[54]. Earlier reports from the UK showed increased mortality among ethnic minorities where the exact explanations were not clear, but it was believed to be due to biological, medical, or sociological factors[55,56]. Given the known risk factors for COVID-19 complications, the confluence of hypertension, diabetes, obesity, and the higher prevalence of a cardiovascular disease among black persons may be driving these early signals. Among the two most populous countries in Africa, Nigeria and Ethiopia, the national prevalence of hypertension is 28.9% and 19.6% and diabetes mellitus is 5.77% and 5%, respectively[57,11,58,59]. And where there is a lack of awareness and poor health-seeking behavior in our population, many patients present with uncontrolled diseases with attending complications which could contribute to the severity of their illness[60].

Minor comments:

a)• Authors should rewrite the results section to ensure the result section looks well connected and free from any grammatical errors.

 Response: Thank you for the comment. We edited the grammatical errors and tried to give it a better flow (as well illustrated in track changes).

b)• Please sure that ethical protocol approval number and certificate should be incorporated in the supplementary data alongside a sample consent form approved by the ethical committee.

 Response: We incorporated the protocol number in the main text Methods section and the supplementary information section. But for the consent form due to the retrospective nature of our study, we were granted a waiver and we didn’t use one. 

Reviewer #2: The data presented in the manuscript is collected from the patients admitted with severe/non-severe COVID-19 disease at the Eka Kotebe General Hospital, COVID-19 Isolation and Treatment Center. The treatment regimen remained standard within the study group. The manuscript is well written with statistically significant evaluation of the comorbidities associated with COVID-19.

 Response: Thank you for the comment.

Reviewer #3: This is an association study between different risk factors with the severity of COVID-19 pneumonia in patients. The study included 265 severe COVID-19 and 190 with the non-severe disease. While the data is interesting, the manuscript has significant omissions. My primary concerns are the following:

-1) Recent publication from Leulseged et al, 2022 “COVID-19 disease severity and associated factors among Ethiopian patients: A study of the millennium COVID-19 care center” have performed similar study in Ethiopian population in 686 patients. Can authors address how their study is different from this recent publication?

 Response: Thanks for the concern raised. 

We are aware of the recent report from the Millennium Center, which has conclusions that are similar to our study. However, we feel that our paper adds to the literature in a significant manner. Ethiopia is the second-largest country in Africa by population, with some 115 million people, and a single report from a single center simply cannot suffice to describe any aspect of Covid-19 completely. Our study reports a cohort independent from the prior study, with a different methodology. Even if the results are somewhat similar, these are valuable and novel data.

Our study aimed to show mainly the association between hypertension and severity of COVID-19 pneumonia, and we also assessed the relation with other comorbidities, clinical, and laboratory characteristics of patients admitted at Eka General Hospital, which was the first and the only hospital in the capital dedicated as a whole for COVID-19 patients isolation and treatment and it is still the only center that provides dialysis/renal replacement therapy, surgical, obstetrics and gynecologic care for COVID-19 positive patients which makes it the main center for referral and admission of cases who require advanced or critical care. We compared differences between those with non-severe (mild/moderate COVID-19) to those with severe (severe/critical COVID-19) diseases, which we considered a matched case-control design to show an association. Whereas, Leulseged et al, 2022 study assessed COVID-19 patients admitted to Millunium COVID-19 Care Center (MCCC), the other care center in the capital which was a repurposed field hospital from a multipurpose recreation/exhibition center, for any characteristics associated with COVID-19 severity. They conducted a cross-sectional study on all selected patients of all age groups. Regarding ours, it was a matched case-control study, where cases were all adult RT-PCR confirmed COVID-19 patients with severe/critical disease. At the same time, controls were those age and gender-matched RT-PCR confirmed COVID-19 patients with mild/moderate disease who were discharged within the study period.

Furthermore, the other difference was sample size determination and sampling technique, they determined it using a double population proportion formula with the assumptions of; a 95% confidence interval, power of 80%, the proportion of males who had the severe disease as 80%, the proportion of females who had the severe disease as 75% and considering a non-response rate of 10%, using a simple random sampling technique. In our study, the sample size determination was based on the assumption of the two-sided significance level of a 95%, power of 80%, control to case ratio of 3 to 1, taking a proportion of hypertension among the controls to be 19.6% (inferred from the population-based study) and with the assumption of the proportion to be double among the cases (39.2%). The study included all cases sequentially, and the next five consecutive non-cases for each case were included (which were age and gender-matched).

Despite the above difference, we learned a lot from their study and tried to improve some aspects of our association study. Notably, we decided to match the participants for important demographic characteristics and to omit clinical characteristics that had pathobiological or effect-modifying relation with a severe manifestation of a disease (in our case we omitted such potential modifiers after checking for bivariate collerate and multicollinear effect).

In our study, the association assessment was focused on a specific factor or variable mainly (hypertension) as an entry with patients grouped as severe and non-severe, whereas the Leulseged et al. 2022 tried to show association assessment with several factors across the severity spectrum of COVID-19 (mild, moderate, severe) separately in addition to the difference in the study population, sample size determination, and sampling technique.

- 2)The key problem with this study is that in the complex phenotype field, few researchers would have confidence in an association that is based on these small sample sizes. (Even if the authors stated as the power of the study).

 Response: We agreed with the reviewer and here we tried to match the two study arms with important moderators(discharge date, age, and gender) of the outcome. But still, it will be complex if we consider matching for comorbidity or substance use, or others. For the novelty of the evidence to be generated, we opted for a 1:3 ratio case/control to improve the power of the study with strict sampling techniques.

- 3)The discussion could be made much more interesting. As it is, they primarily reiterate their results and provide a bit more literature review. However, it would be interesting if they could discuss clinical implications and areas for further study.

 Response: This is an important comment and we incorporated pathological features and clinical implications with areas for further study in the Discussion section.

In paragraph 7 of the Discussion section:

 The explanation for the higher prevalence of hypertension, diabetes mellitus, and cardiac disease among COVID-19 patients may focus on the SARS-CoV-2 cell entry mechanism. Similar to SARS-CoV-1, SARS-CoV-2 contains a receptor-binding domain (RBD) that recognizes angiotensin-converting enzyme 2 (ACE2) as its receptor with a higher binding affinity compared to SARS-CoV-1 [32]. Epithelial cells of the lungs, intestine, kidney, and blood vessels are identified to have abundant ACE2 receptors [33]. Hence, increased expression of ACE2 may promote the internalization of SARS-CoV-2, which in turn may increase the chances of developing COVID-19 or a severe form of the disease[34]. The severity of COVID-19 among patients with hypertension, diabetes, and chronic cardiac disease also may be partially explained by the increased incidence of thrombotic complications which is a fact that patients with these conditions are at increased risk of thrombotic events [35,36]. These chronic medical conditions often present with inflammation and weakened innate immune responses in affected individuals. This may predispose those individuals to infections and disease complications [37]. The high prevalence of fatal cases among COVID-19 patients with hypertension, diabetes, and chronic cardiac disease as comorbidity could be due to the induction of cytokine storm. Cytokine storms resulting in hyper-inflammation are the hallmarks of severe SARS-CoV-2 infection [38]. Metabolic inflammation as a consequence of hypertension, diabetes, and chronic cardiac disease is also known to compromise the immune system. These patients are commonly reported to have weakened immunological function arising from reduced macrophage and lymphocyte activity which could predispose individuals to infections, especially those infections for which cell-mediated immunity constitutes an important host defense [39].

---

## [Decision Letter · Decision Letter 1]

12 May 2022

PONE-D-22-04541R1Assessment of hypertension and other factors associated with the severity of disease in COVID-19 pneumonia, Addis Ababa, Ethiopia: A case-control studyPLOS ONE

Dear Dr. Ashamo,

Thank you for submitting your manuscript to PLOS ONE. After careful consideration, we feel that it has merit but does not fully meet PLOS ONE’s publication criteria as it currently stands. Therefore, we invite you to submit a revised version of the manuscript that addresses the points raised during the review process.

We look forward to receiving your revised manuscript.

Kind regards,

Kanhaiya Singh, Ph.D

Academic Editor

PLOS ONE

Additional Editor Comments (if provided):

Please address to the concerns raised by Reviewer 1. In-depth data analysis and presentation is necessary.

Reviewers' comments:

Reviewer's Responses to Questions

**Comments to the Author**

1. If the authors have adequately addressed your comments raised in a previous round of review and you feel that this manuscript is now acceptable for publication, you may indicate that here to bypass the “Comments to the Author” section, enter your conflict of interest statement in the “Confidential to Editor” section, and submit your "Accept" recommendation.

Reviewer #1: (No Response)

Reviewer #3: All comments have been addressed

2. Is the manuscript technically sound, and do the data support the conclusions?

Reviewer #1: No

Reviewer #3: Yes

3. Has the statistical analysis been performed appropriately and rigorously? 

Reviewer #1: No

Reviewer #3: Yes

4. Have the authors made all data underlying the findings in their manuscript fully available?

Reviewer #1: No

Reviewer #3: Yes

5. Is the manuscript presented in an intelligible fashion and written in standard English?

Reviewer #1: No

Reviewer #3: Yes

6. Review Comments to the Author

Reviewer #1: This manuscript entitled "Assessment of hypertension and other factors associated with the severity of disease

in COVID-19 pneumonia" has been thoroughly assessed based on the overall weakness and strength of the manuscript. The manuscript hold scientific scope and can contribute in the unveiling pathogenesis of COVID19. However, after in depth analysis of manuscript a few significant inferences were noted.

1. The results appear to be too preliminary and disjointed for publication.

2. The authors have not enriched the manuscript significantly following the previous peer reviewed comments.

3. The authors should work on data presentation and in-depth analysis of data alongside expansion of the cohort size to meet the publication quality of the journal.

Abiding following points i would urge the editorial committee not to further consider this manuscript for publication in present format.

Reviewer #3: Thank you for the response. The authors have addressed my concerns.

7. PLOS authors have the option to publish the peer review history of their article (what does this mean?). If published, this will include your full peer review and any attached files.

Reviewer #1: No

Reviewer #3: No

---

## [Author Response · Author response to Decision Letter 1]

24 Jun 2022

PLOS ONE

June 22, 2022

Dear Editor,

We greatly appreciate the comments of the reviewers of our manuscript and have addressed them here, point by point, and revised our manuscript accordingly.

Reviewer #1: Work on data presentation and in-depth analysis of data alongside an expansion of the cohort size to meet the publication quality of the journal

Response: Thank you for the comments and concerns raised to perform deep analysis using Cox proportion hazard (CoH). Here, I will try to kindly respond to why we couldn’t conduct the analysis further. As you may recall, our study design was a case-control where recruitment of study subjects was based on the severity of disease (COVID-19) status. Cases were those with severe forms of the disease including critical, while controls were those who had mild to moderate disease, including asymptomatic ones. Some of the cases were recruited when they came to the hospital in a severe or critical state, while the rest were recruited after they developed severe disease. The exposure was hypertension whereas other related factors were considered possible confounders to suppress the effect of the main exposure(hypertension). Most importantly, from the nature of our retrospective data, the time between the onset of the disease and the change in severity course is difficult to retrieve and estimate. Thus, I respectfully would like to notify you that the design of our study will not allow us to use the CoH. 

Concerning the sample size, we edited the sentence under the strength and limitation part, under the discussion section, where we tried to clarify for you the points we raised about the disparity between the initial calculated sample size (252, composed of 63 cases and 189 controls) and the actual number of patients (265, composed of 75 cases and 190 controls) with majority matched 1 to 3. So, here we acknowledge the limitation we faced due to the strict matching criteria of the design, and we were short of controls matching to cases for age bands, sex, and discharge date criteria though we had an extra number of cases (75 vs 63). 

In the paragraph on Strength and limitation part of the Discussion section

However, this study had a few limitations. First, the strict matching criteria, including the date of discharge on selecting one case for three controls, failed to fulfill the planned case-to-control ratio, we collected data for 75 cases for which we were supposed to collect proportionally 225 controls which makes the total sample size to be 300. However, we were able to retrieve data of 75 cases for 190 controls which made the total sample size 265, which is a bit higher than the initial calculated sample size (252, 63 cases for 189 controls) with a slightly higher number of those on case arm (75 vs 63).

Thank you for considering our research article. We hope that our revisions meet with your approval.

With best regards,

---

## [Editor Report · Decision Letter 2]

1 Aug 2022

Assessment of hypertension and other factors associated with the severity of disease in COVID-19 pneumonia, Addis Ababa, Ethiopia: A case-control study

PONE-D-22-04541R2

Dear Dr. Ashamo,

We’re pleased to inform you that your manuscript has been judged scientifically suitable for publication and will be formally accepted for publication once it meets all outstanding technical requirements.

Kind regards,

Kanhaiya Singh, Ph.D

Academic Editor

PLOS ONE
---

## [Editor Report · Acceptance letter]

5 Aug 2022

PONE-D-22-04541R2 

Assessment of hypertension and other factors associated with the severity of disease in COVID-19 pneumonia, Addis Ababa, Ethiopia: A case-control study 

Dear Dr. Ashamo:

I'm pleased to inform you that your manuscript has been deemed suitable for publication in PLOS ONE. Congratulations! Your manuscript is now with our production department. 

Kind regards, 

on behalf of

Dr. Kanhaiya Singh 

Academic Editor

PLOS ONE